# UniSDF: Unifying Neural Representations for High-Fidelity 3D Reconstruction of Complex Scenes with Reflections

**Fangjinhua Wang**[*]
ETH Zurich

**Marie-Julie Rakotosaona**
Google

**Michael Niemeyer**
Google

**Richard Szeliski**
Google

**Marc Pollefeys**
ETH Zurich

**Federico Tombari**
Google

## Abstract

Neural 3D scene representations have shown great potential for 3D reconstruction from 2D images. However, reconstructing real-world captures of complex scenes still remains a challenge. Existing generic 3D reconstruction methods often struggle to represent fine geometric details and do not adequately model reflective surfaces of large-scale scenes. Techniques that explicitly focus on reflective surfaces can model complex and detailed reflections by exploiting better reflection parameterizations. However, we observe that these methods are often not robust in real scenarios where non-reflective as well as reflective components are present. In this work, we propose UniSDF, a general purpose 3D reconstruction method that can reconstruct large complex scenes with reflections. We investigate both camera view as well as reflected view-based color parameterization techniques and find that explicitly blending these representations in 3D space enables reconstruction of surfaces that are more geometrically accurate, especially for reflective surfaces. We further combine this representation with a multi-resolution grid backbone that is trained in a coarse-to-fine manner, enabling faster reconstructions than prior methods. Extensive experiments on object-level datasets DTU, Shiny Blender as well as unbounded datasets Mip-NeRF 360 and Ref-NeRF real demonstrate that our method is able to robustly reconstruct complex large-scale scenes with fine details and reflective surfaces, leading to the best overall performance. Project page: https://fangjinhuawang.github.io/UniSDF.

## 1 Introduction

Given multiple images of a scene, accurately reconstructing a 3D scene is an open problem in 3D computer vision. 3D meshes from reconstruction methods can be used in many downstream applications, *e.g.*, scene understanding, robotics, and creating 3D experiences for augmented/virtual reality [36, 51]. Typical aspects of real-world scenes such as uniformly colored areas or non-Lambertian surfaces remain challenging.

As a traditional line of research, multi-view stereo methods [39, 49, 16, 46] usually estimate depth maps with photometric consistency and then reconstruct the surface as a post-processing step, *e.g.*, point cloud fusion with screened Poisson surface reconstruction [19] or TSDF fusion [10]. However, they cannot reconstruct reflective surfaces since their appearances are not multi-view consistent.

---

[*]This work was conducted during an internship at Google.

38th Conference on Neural Information Processing Systems (NeurIPS 2024).

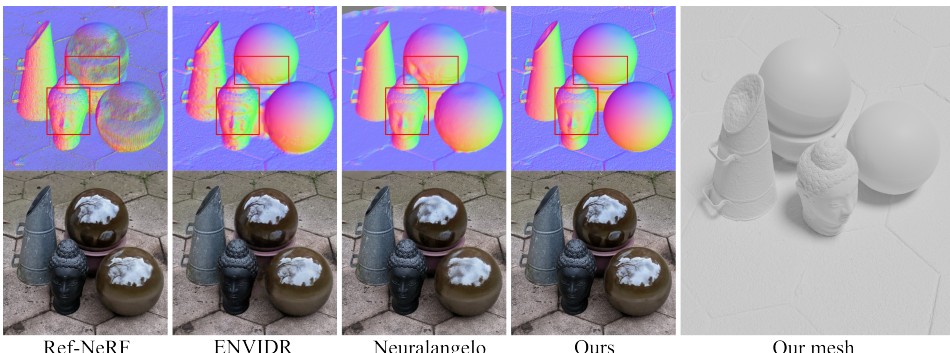

| Ref-NeRF | ENVIDR | Neuralangelo | Ours | Our mesh |

Figure 1: Comparison of surface normals (top) and RGB renderings (bottom) on "garden spheres" [44]. While the state-of-the-art methods Ref-NeRF [44], ENVIDR [22], and Neuralangelo [21] struggle to reconstruct reflective elements or fine geometric details, our method accurately models both, leading to high-quality mesh reconstructions of all parts of the scene. Best viewed when zoomed in.

Recently, Neural Radiance Fields (NeRF) [28] render compelling photo-realistic images by parameterizing a scene as a continuous function of radiance and volume density using a multi-layer perceptron (MLP). More recent works [30, 8, 42, 4] replace or augment MLPs with grid based data structures to accelerate training. For example, Instant-NGP (iNGP) [30] uses a pyramid of grids and hashes to encode features and a tiny MLP to process them. Motivated by NeRF, neural implicit reconstruction methods [50, 47] combine signed distance functions (SDF) with volume rendering, and produce smooth and complete surfaces. For acceleration, recent works [21, 37] rely on hash grid representations and reconstruct surfaces with finer details. However, these NeRF-based methods cannot accurately reconstruct reflective surfaces [44, 14].

To better represent the reflective appearance, Ref-NeRF [44] parameterizes the appearance using *reflected* view direction that exploits the surface normals, while NeRF uses the camera view direction. Recently, some works [51, 22, 25, 14] adopt this reflected view parameterization and successfully reconstruct reflective surfaces. We observe that while reflected view radiance fields can effectively reconstruct highly specular reflections, they struggle to represent more diffuse or ambiguous reflection types and fine details that can be found in real scenes. In contrast, we find that direct camera view radiance fields are more robust to difficult surfaces in real settings, although the reconstructions still present artifacts for reflective scenes. In this paper, we seamlessly bring together reflected view and camera view radiance fields into a novel unified radiance field for representing 3D real scenes accurately in the presence of reflections. Our method is robust for reconstructing both real challenging scenes and highly reflective surfaces.

The proposed method, named UniSDF, performs superior to or on par with respective state-of-the-art methods which are tailored for a specific scene type. UniSDF can be applied to any type of dataset, ranging from DTU [1], Shiny Blender [44], Mip-NeRF 360 dataset [3], to Ref-NeRF real dataset [44], leading to the overall best performance. It demonstrates the capability to accurately reconstruct complex scenes with large scale, fine details and reflective surfaces as we see in Fig. 1.

In summary, we propose a novel algorithm that learns to seamlessly combine two radiance fields with a learnable weight field while exploiting the advantages of each representation. Our method produces high quality surfaces in both reflective and non-reflective regions.

## 2    Related Works

**Multi-view stereo (MVS).**   Many traditional [39, 48] and learning-based [49, 16, 46, 45] MVS methods first estimate multi-view depth maps and then reconstruct the surface by fusing depth maps in a post-processing step. As the core step, depth estimation is mainly based on the photometric consistency assumption across multiple views. However, this assumption fails for glossy surfaces with reflections, and thus MVS methods cannot reconstruct them accurately.

**Neural radiance fields (NeRF).** As a seminal method in view synthesis, NeRF [28] represents a scene as a continuous volumetric field with an MLP, with position and camera view direction as inputs, and renders an image using volumetric ray-tracing. Since NeRF is slow to train, some methods [30, 42, 8] use voxel-grid-like data structures to accelerate training. Many follow-up works apply NeRFs to different tasks, *e.g.*, sparse-view synthesis [54, 31, 43], real-time rendering [9, 34, 18, 53], 3D generation [33, 23, 7] and pose estimation [24, 41, 59]. For the 3D reconstruction task, there are many methods [32, 50, 47, 55, 13, 21, 37, 26, 35] integrating NeRF with signed distance functions, a common implicit function for geometry. Specifically, they transform SDFs back to volume density for volume rendering. However, we observe that they are unable to reconstruct shiny / reflective surfaces since NeRF's camera view direction parameterization for the color prediction does not accurately model reflective parts of the scene.

**NeRFs for reflections.** To render reflective appearance, [40, 5, 57, 58] extend NeRF and decompose a scene into physical components with strong simplifying assumptions, *e.g.*, known lighting [40] or no self-occlusion [5, 57]. Recently, Ref-NeRF [44] reparameterizes the appearance prediction with separate diffuse and reflective components by using the reflected view direction, which improves the rendering of specular surfaces. As a result, recent works [51, 22, 25, 14, 27] adopt this representation to reconstruct glossy surfaces. While leading to strong view-synthesis for reflections, we find that reflected view radiance field approaches often lead to overly smooth reconstructions with missing details and that their optimization is not stable on real-world scenes. In contrast to existing methods with a single radiance field, we propose to seamlessly combine reflected view and camera view radiance fields into a novel unified radiance field with learnable weight, which is robust for reconstruction in challenging scenes with reflective surfaces. The recent preprint Factored-NeuS [12] also uses camera view and reflected view radiance fields. It separately supervises the rendered colors of two radiance fields with ground-truth color, instead of learning a weight field to combine them like ours. We find that our approach to combine two radiance fields with learnable weight is simpler to train and leads to better reconstruction. Other recent methods [17, 56, 52] use weight to compose the colors from camera view radiance field(s) to render reflections, similarly to us. [17, 56] can only handle planar reflections, *e.g.*, mirrors, and require ground-truth masks of reflective objects to supervise the weight. In contrast, our method can handle non-planar reflective objects, *e.g.*, spheres. As we learn the weight to compose camera view and reflected view radiance fields *without supervision*, our method does not require additional input and can be trained from only RGB images. MS-NeRF [52] uses multiple volume density fields and how to reconstruct the underlying surface is undefined. In contrast, we use a single SDF field to represent geometry and can hence directly extract the iso-surface from it.

# 3 Method

In this section, we first review the basic elements of NeRF [28]. We then describe the architecture and training strategy of our method.

## 3.1 NeRF Preliminaries

In NeRF [28], a 3D scene is represented by mapping a position $\mathbf{x}$ and ray direction $\mathbf{d}$ to a volumetric density $\sigma$ and color $\mathbf{c}$ using MLP. For a pixel in the target viewpoint and its corresponding ray $\mathbf{r} = \mathbf{o} + t\mathbf{d}$, distance values $t_i$ are sampled along the ray. The density $\sigma_i$ is predicted by a spatial MLP that receives the position $\mathbf{x}$ as input, while the directional MLP that predicts the color $\mathbf{c}_i$ uses the bottleneck vector $\mathbf{b}(\mathbf{x})$ from the density MLP and the view direction $\mathbf{d}$ as input. The final color $\mathbf{C}$ is rendered as:

$$\mathbf{C} = \sum_i w_i \mathbf{c}_i, w_i = T_i \alpha_i, \tag{1}$$

where $\alpha_i = 1 - \exp(-\sigma_i \delta_i)$ is opacity, $\delta_i = t_i - t_{i-1}$ is the distance between adjacent samples, and $T_i = \prod_{j=1}^{i-1}(1 - \alpha_j)$ is the accumulated transmittance. The model is trained by minimizing the loss between the predicted and ground truth color:

$$\mathcal{L}_{\text{color}} = \mathbb{E}[(||\mathbf{C} - \mathbf{C}_{gt}||^2]. \tag{2}$$

Note that Mildenhall *et al.* [28] use a single-layer directional MLP and thus often describe the combination of NeRF's spatial and view dependence MLPs as a single MLP.

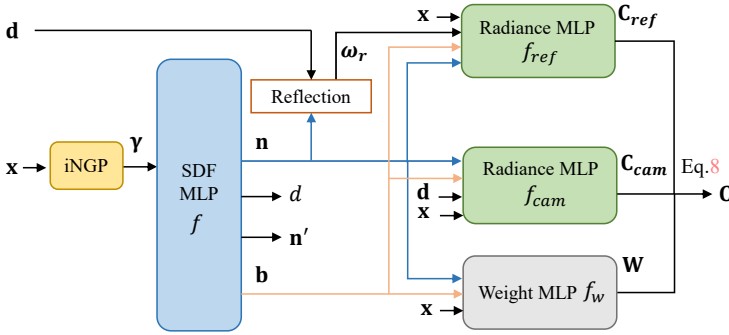

Figure 2: Pipeline of UniSDF. We combine the camera view radiance field and reflected view radiance field in 3D. Given a position $\mathbf{x}$, we extract iNGP features $\gamma$ and input them to an MLP $f$ that estimates a signed distance value $d$ used to compute the NeRF density. We parametrize the camera view and reflected view radiance fields with two different MLPs $f_{cam}$ and $f_{ref}$ respectively. Finally, we learn a continuous weight field that is used to compute the final color as a weighted composite $\mathbf{W}$ of the radiance fields colors $\mathbf{C}_{cam}$ and $\mathbf{C}_{ref}$ after volume rendering, Eq. 8.

## 3.2 UniSDF

Given a set of known images of a scene that potentially contains reflective surfaces, our goal is to optimize a neural implicit field and reconstruct the scene with high fidelity and geometric accuracy. We propose UniSDF, a method that enables us to seamlessly combine camera view radiance fields and reflected view radiance fields to reconstruct both (a) non-reflective surfaces, diffuse reflective surfaces and complex surfaces with both reflective and non-reflective areas as well as (b) highly specular surfaces with a well defined and detailed reflected environment. Our pipeline is shown in Fig. 2. We generate two radiance fields that are parameterized by camera view directions or reflected view directions and combine them at the pixel level using a learned rendered weight.

**Volume rendering the SDF.** We represent the scene geometry using a signed distance field (SDF), which defines the surface $\mathcal{S}$ as the zero level set of SDF $d$:

$$\mathcal{S} = \{\mathbf{x} : d(\mathbf{x}) = 0\}. \tag{3}$$

To better reconstruct large-scale scenes, we follow Mip-NeRF 360 [3] and transform $\mathbf{x}$ into a *contracted space* with the following contraction:

$$\text{contract}(\mathbf{x}) = \begin{cases} \mathbf{x} & ||\mathbf{x}|| \leq 1 \\ \left(2 - \frac{1}{||\mathbf{x}||}\right)\left(\frac{\mathbf{x}}{||\mathbf{x}||}\right) & ||\mathbf{x}|| > 1 \end{cases} \tag{4}$$

For volume rendering, we compute the volume density $\sigma(\mathbf{x})$ from the signed distance $d(\mathbf{x})$ as: $\sigma(\mathbf{x}) = \alpha\Psi_\beta(d(\mathbf{x}))$, where $\Psi_\beta$ is the cumulative distribution function of a zero-mean Laplace distribution with learnable scale parameter $\beta > 0$. The surface normal at $\mathbf{x}$ can be computed as the gradient of the signed distance field: $\mathbf{n} = \nabla d(\mathbf{x})/||\nabla d(\mathbf{x})||$.

**Hash Encoding with iNGP.** To accelerate training and improve reconstruction of high-frequency details, we use iNGP [30] to map each position $\mathbf{x}$ to a higher-dimensional feature space. Specifically, the features $\{\gamma_l(\mathbf{x})\}$ from the pyramid levels of iNGP are extracted with trilinear interpolation and then concatenated to form one single feature vector $\gamma(\mathbf{x})$, which is passed to the SDF MLP.

**Camera View & Reflected View Radiance Fields.** In contrast to most existing methods [28, 2, 44] that use a single radiance field, we propose to combine a camera view radiance field and a reflected view radiance field to better represent reflective and non reflective surfaces.

We follow NeRF [28] for representing our camera view radiance field $\mathbf{c}_{cam}$, which is computed from features defined at each position and the camera view direction:

$$\mathbf{c}_{cam} = f_{cam}(\mathbf{x}, \mathbf{d}, \mathbf{n}, \mathbf{b}), \tag{5}$$

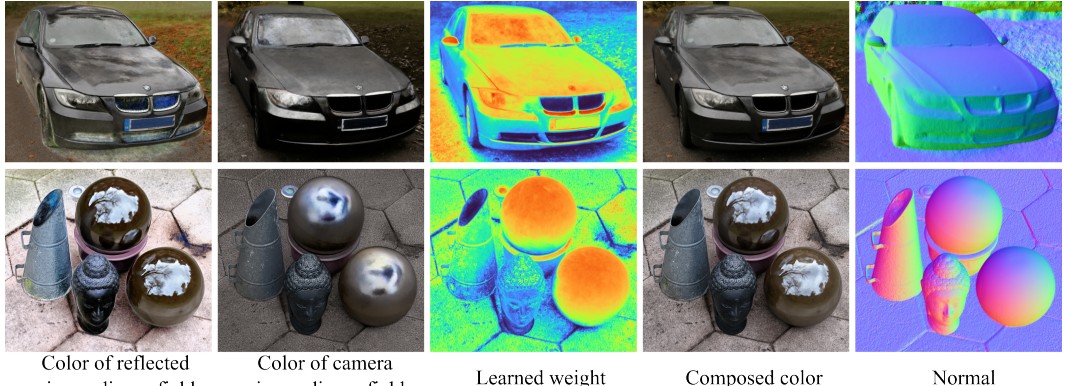

| Color of reflected view radiance field | Color of camera view radiance field | Learned weight | Composed color | Normal |

Figure 3: Visualization of the color of reflected view radiance field, color of camera view radiance field, learned weight $\mathbf{W}$, composed color and surface normal on "sedan" and "garden spheres" scenes [44]. Our method assigns high weight (red color) for reflective surfaces, *e.g.*, window and hood of sedan, spheres, without any supervision.

where $\mathbf{b}$ is the bottleneck feature vector from SDF MLP, $\mathbf{n}$ is the normal at $\mathbf{x}$ and $\mathbf{d}$ is the camera view direction. Similarly to recent works [50, 47], we notice that using surface normals as input leads to better quality.

We represent the reflected radiance field $\mathbf{c}_{ref}$ with an MLP $f_{ref}$ as:

$$\mathbf{c}_{ref} = f_{ref}(\mathbf{x}, \omega_r, \mathbf{n}, \mathbf{b}), \tag{6}$$

where $\omega_r$ is the reflected view direction around the normal $\mathbf{n}$. In Ref-NeRF [44], it is shown that for BRDFs under a limited set of conditions, view-dependent radiance is a function of $\omega_r$ only. Unlike Ref-NeRF, which uses separate diffuse and specular components, we only use the specular component, leading to a simpler architecture. Additionally, we observe that using separate diffuse and specular components can lead to optimization instabilities resulting in geometry artifacts (see supp. mat. for details).

The main difference between two radiance fields is the view directional input of the MLP. As shown in Fig. 3, our method mainly uses the reflected view radiance field to represent highly specular reflections such as the tree reflections in the garden spheres or the environment reflection on the sedan car. The camera view radiance field is used to represent more diffuse reflections.

**Learned composition.** We compose two radiance fields using a learnable weight field in 3D. Specifically, we use an MLP $f_w$ to learn the weight values $\mathbf{w}$:

$$\mathbf{w} = \text{sigmoid}\left(f_w(\mathbf{x}, \mathbf{n}, \mathbf{b})\right). \tag{7}$$

We compose the signals at the pixel level. We first volume render $\mathbf{W}, \mathbf{C}_{ref}, \mathbf{C}_{cam}$ following Eq. 1. We then compose the colors for each pixel as follows:

$$\mathbf{C} = \mathbf{W} \cdot \mathbf{C}_{ref} + (1 - \mathbf{W}) \cdot \mathbf{C}_{cam}. \tag{8}$$

In Fig. 3, the weight $\mathbf{W}$ detects reflections well and assigns high weight to reflected view radiance field in reflective regions. The surface normals show that our model accurately reconstructs both reflective and non-reflective surface geometry.

**Motivation of composing radiance fields.** Disambiguating the influence of geometry, color and reflection is an ill-posed problem in 3D reconstruction from images. NeRF-based methods [47, 50, 21] with camera view radiance field show their robustness in real-world scenes [1], while having difficulty with reflections [44, 14]. Ref-NeRF based methods [25, 14, 51] with reflected view radiance field usually perform well under restricted conditions, *e.g.*, highly specular objects [44], while we experimentally find their performance degrades in real-world scenes, *e.g.*, NeRO [25] and Ref-NeuS [14] on DTU [1] (Tab. 1), BakedSDF [51] on Mip-NeRF 360 dataset [3] (Fig. 4). Therefore, to extend theoretically justified Ref-NeRF representation with robust scene representations,

we propose to exploit the advantages of two radiance fields by combining them with learnable weight. Moreover, since each type of radiance field is specialized for different levels of reflection strength and complexity, we observe that the reconstructed geometries while using the two types of radiance are often complementary (Fig. 5). In our method, we explicitly intertwine the radiance fields in 3D to continuously determine and use the most adapted parametrization for each surface area.

## 3.3 Training and Regularization

**Coarse-to-fine training.** We observe that directly optimizing all the features in our multi-resolution hash grid leads to overfitting of training images, in particular to specular appearance details, which in turn results in incorrect geometry as we show in Fig. 7 (a). We observe that this model tends to fake specular effects by embedding emitters inside the surface exploiting the numerous learnable features in the hash grid. Therefore, we propose to instead optimize the hash grid features in a coarse-to-fine fashion, similarly to [21, 37], to avoid overfitting and promote smoother and more realistic surfaces. Specifically, we start with $L_{\text{init}}$ coarse pyramid levels in the beginning of training, and introduce a new level with higher resolution every $T_0$ training fraction (see implementation details in Sec. 4.1).

**Regularization.** Following prior works [50, 47], we use an eikonal loss [15] to encourage $d(\mathbf{x})$ to approximate a valid SDF:

$$\mathcal{L}_{\text{eik}} = \mathbb{E}_{\mathbf{x}}[(||\nabla d(\mathbf{x})|| - 1)^2]. \tag{9}$$

To promote normal smoothness, we constrain the computed surface normal $\mathbf{n}$ to be close to a predicted normal vector $\mathbf{n}'$. $\mathbf{n}'$ is predicted by the SDF MLP and normalized. We use the normal smoothness loss $\mathcal{L}_{\text{p}}$ [44] as:

$$\mathcal{L}_{\text{p}} = \sum_i w_i ||\mathbf{n} - \mathbf{n}'||^2. \tag{10}$$

We also use the orientation loss $\mathcal{L}_{\text{o}}$ from Ref-NeRF [44] to penalize normals that are "back-facing", using:

$$\mathcal{L}_{\text{o}} = \sum_i w_i \max(0, \mathbf{n} \cdot \mathbf{d})^2. \tag{11}$$

**Full loss function.** The full loss function $\mathcal{L}$ includes the color loss $\mathcal{L}_{\text{color}}$ of composed color $\mathbf{C}$ and the regularizations, which is written as follows:

$$\mathcal{L} = \mathcal{L}_{\text{color}} + \lambda_1 \mathcal{L}_{\text{eik}} + \lambda_2 \mathcal{L}_{\text{p}} + \lambda_3 \mathcal{L}_{\text{o}}. \tag{12}$$

# 4 Experiments

## 4.1 Experimental Settings

**Datasets.** We extensively evaluate our method on four different types of datasets. The DTU dataset [1] is an indoor object-centric dataset with ground truth point clouds. Following prior works [50, 47], we use the same 15 scenes for evaluation. The Shiny Blender dataset [44] contains six different shiny objects that are rendered in Blender under conditions similar to the NeRF dataset. The Mip-NeRF 360 dataset is proposed in [3] and contains complex unbounded indoor and outdoor scenes captured from many viewing angles. We further evaluate on the three large-scale scenes with reflections that are introduced in Ref-NeRF [44], which consists of the scenes "sedan", "garden spheres" and "toycar". For simplicity, we name these 3 scenes the "Ref-NeRF real dataset".

**Implementation details.** Based on the Mip-NeRF 360 codebase [29], we implement our method in Jax [6] with the re-implementation of VolSDF [50] and iNGP [30]. In our iNGP hierarchy of grids and hashes, we use 15 levels from 32 to 4096, where each level has 4 channels. For coarse to fine training, we set $L_{\text{init}} = 4$ and $T_0 = 2\%$. Similar to mip-NeRF 360 [3], we use two rounds of proposal sampling and then a final NeRF sampling round. Following Zip-NeRF, we penalize the sum of the mean of squared grid/hash values at each pyramid level with a loss multiplier as 0.1. Our models are all trained on 8 NVIDIA Tesla V100-SXM2-16GB GPUs with a batch size of $2^{14}$. We train 25k steps on DTU / Shiny Blender and 100k steps on Mip-NeRF 360 / Ref-NeRF real datasets, which takes 0.75h and 3.50h respectively. See the supplement for more details.

Table 1: Quantitative results of Chamfer Distance (C.D.) on DTU [1]. Red, orange and yellow indicate the first, second and third best methods. †: Factored-NeuS [12] does not provide result for scan 69. Its result is the average error of the other 14 scenes.

| Methods | NeuS [47] | NeuralWarp [11] | Geo-NeuS [13] | Neuralangelo [21] |
|---|---|---|---|---|
| C.D. (mm) ↓ | 0.87 | 0.68 | 0.51 | 1.07 |

| Methods | NERO [25] | Ref-NeuS [14] | Factored-NeuS† [12] | Ours |
|---|---|---|---|---|
| C.D. (mm) ↓ | 1.04 | 1.93 | 0.77 | 0.64 |

Table 2: Quantitative results on Shiny Blender [44], Mip-NeRF 360 dataset [3] and Ref-NeRF real dataset [44]. 'Mean' represents the *average* rendering metrics on all datasets. Red, orange, and yellow indicate the first, second, and third best methods for each metric. *: We follow Ref-NeuS [14] and evaluate *accuracy* of mesh on four scenes (car, helmet, toaster, coffee). See supp. mat. for details.

| Methods | Shiny Blender | | | | | Mip-NeRF 360 dataset | | | Ref-NeRF real dataset | | | **Mean** | | |
|---|---|---|---|---|---|---|---|---|---|---|---|---|---|---|
| | PSNR ↑ | SSIM ↑ | LPIPS ↓ | MAE° ↓ | Acc* ↓ | PSNR ↑ | SSIM ↑ | LPIPS ↓ | PSNR ↑ | SSIM ↑ | LPIPS ↓ | PSNR ↑ | SSIM ↑ | LPIPS ↓ |
| Mip-NeRF 360 [3] | 25.49 | 0.939 | 0.122 | - | - | 27.69 | 0.791 | 0.237 | 24.27 | 0.650 | 0.276 | 25.82 | 0.793 | 0.212 |
| Zip-NeRF [4] | 29.24 | 0.942 | 0.112 | - | - | 28.53 | 0.828 | 0.190 | 23.68 | 0.635 | 0.247 | 27.15 | 0.802 | 0.183 |
| Geo-NeuS [13] | 28.78 | 0.945 | 0.085 | 10.52 | 1.63 | - | - | - | - | - | - | - | - | - |
| Neuralangelo [21] | 30.68 | 0.949 | 0.095 | 14.16 | 1.81 | 25.08 | 0.699 | 0.332 | 23.70 | 0.608 | 0.330 | 26.49 | 0.752 | 0.252 |
| Ref-NeRF [44] | 35.96 | 0.967 | 0.058 | 18.38 | - | - | - | - | 24.06 | 0.589 | 0.355 | - | - | - |
| ENVIDR [22] | 35.85 | 0.983 | 0.036 | 4.61 | - | - | - | - | - | - | - | - | - | - |
| NeRO [25] | 29.84 | 0.962 | 0.072 | - | - | - | - | - | - | - | - | - | - | - |
| Ref-NeuS [14] | 27.40 | 0.951 | 0.073 | 5.34 | 0.85 | - | - | - | - | - | - | - | - | - |
| Factored-NeuS [12] | 30.89 | 0.954 | 0.076 | 5.31 | 1.90 | - | - | - | - | - | - | - | - | - |
| BakedSDF [51] | 25.60 | 0.943 | 0.090 | - | - | 26.42 | 0.738 | 0.314 | 24.43 | 0.636 | 0.325 | 25.48 | 0.772 | 0.243 |
| Ours | 36.82 | 0.976 | 0.043 | 4.76 | 1.06 | 27.67 | 0.808 | 0.213 | 23.70 | 0.636 | 0.265 | 29.40 | 0.807 | 0.174 |

**Baselines.** We compare our method to state-of-the-art volumetric implicit methods in surface reconstruction [47, 11, 13, 21, 51, 25, 14, 22, 12] and view synthesis [3, 44, 4]. Neuralangelo [21] and Zip-NeRF [4] are hash grid-based state-of-the-art methods for reconstruction and view synthesis, respectively. Tailored for handling reflections, [25, 14, 22, 12] are top performing methods for reconstructing and rendering objects with reflective surfaces. BakedSDF [51] is a top performing method for reconstructing high quality mesh of unbounded scenes with reflective surfaces.

To further evaluate the effectiveness of our method, we propose two custom baselines, named "CamV" and "RefV". Using the same backbone as our method, "CamV" uses only the camera view radiance field, while "RefV" uses only the reflected view radiance field following Ref-NeRF [44]. Note that for both baselines, we also use our coarse-to-fine training strategy to improve performance.

## 4.2 Evaluation Results

**DTU.** We evaluate the reconstruction quality on DTU dataset [1]. Following prior works, we extract the mesh at 512 resolution. For Neuralangelo [21], we report the reproduced results from the official implementation. As shown in Tab. 1, Geo-NeuS [13] performs best on DTU and our method outperforms the remaining methods. Geo-NeuS heavily relies on supervision from accurate SfM point cloud and photometric consistency constraint to achieve top performance, while our method uses rendering loss only like Neuralangelo [21]. For reflective regions, the SfM reconstruction is inaccurate for supervision [14] and the multi-view photometric consistency is not guaranteed. We show that Geo-NeuS performs worse on Shiny Blender [44] in Tab. 2. Besides, we observe that NERO [25] and Ref-NeuS [14] perform worse than their baseline NeuS [47]. Though these methods perform well on objects with strong reflections, *e.g.*, Shiny Blender [44], they are not robust in general real-world scenes without strong reflections.

**Shiny Blender.** We summarize the rendering metrics, mean angular error (MAE) of normals and *accuracy* (Acc) of mesh in Tab. 2. Our method performs best in PSNR and on par with ENVIDR [22] in SSIM, LPIPS and MAE. Note that ENVIDR additionally uses an environment MLP, which we do not require, to improve rendering and reconstruction. Besides, we find ENVIDR unrobust in real-world scenes with geometry artifacts on both reflective and non-reflective surfaces, shown in Fig. 1. For mesh quality, our method performs on par with Ref-NeuS [14] in Acc, while performing much better than it on DTU [1] (Tab. 1). This demonstrates the roubustness of our method in various types of scenes. Moreover, our method explicitly outperforms Geo-NeuS [13], Neuralangelo [21] and Zip-NeRF [4] in all metrics.

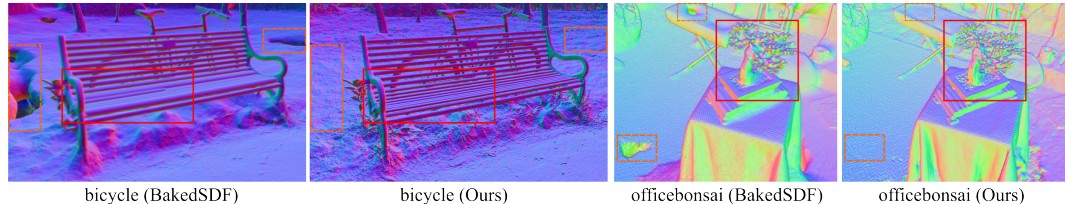

| bicycle (BakedSDF) | bicycle (Ours) | officebonsai (BakedSDF) | officebonsai (Ours) |

Figure 4: Qualitative comparison with BakedSDF [51] on "bicycle" and "officebonsai" scenes of Mip-NeRF 360 dataset [3]. BakedSDF produces hole structures in many regions (highlighted with dotted orange boxes) and less details of fine structures (highlighted with red boxes), while our method reconstructs more complete surfaces and better details. Best viewed when zoomed in.

Table 3: Quantitative comparison with two custom baselines. Best results are in bold. *: RefV fails on scan 110 of DTU [1], the reported chamfer distance (C.D.) is the average of other 14 scenes.

| Methods | DTU | Mip-NeRF 360 dataset | | | Ref-NeRF real dataset | | |
|---|---|---|---|---|---|---|---|
| | C.D. (mm) ↓ | PSNR ↑ | SSIM ↑ | LPIPS ↓ | PSNR ↑ | SSIM ↑ | LPIPS ↓ |
| CamV | 0.85 | 27.26 | 0.800 | 0.225 | 23.30 | 0.622 | 0.283 |
| RefV | 0.89* | 26.74 | 0.794 | 0.223 | 23.02 | 0.615 | 0.301 |
| Ours | **0.64** | **27.67** | **0.808** | **0.213** | **23.70** | **0.636** | **0.265** |

**Mip-NeRF 360 dataset.** As shown in Tab. 2, our method performs on par with Zip-NeRF [4] in rendering. Note that Zip-NeRF focuses on view synthesis, while we focus on surface reconstruction. Compared with BakedSDF [51] and Neuralangelo [21], our performance is much better in all metrics. As shown in Fig. 4, our method reconstructs more complete surfaces and better details, while BakedSDF shows hole artifacts and struggles to reconstruct fine geometric details.

**Ref-NeRF real dataset.** As shown in Tab. 2, evaluation on this dataset is challenging for all methods. Our method outperforms Ref-NeRF [44] and Neuralangelo [21] in SSIM and LPIPS, Zip-NeRF [4] in PSNR and SSIM, and BakedSDF [51] in LPIPS. For surface reconstruction, Neuralangelo [21] cannot reconstruct reflective spheres well as shown in Fig. 1, while our method accurately reconstructs the smooth surface of the reflective spheres and the fine details on the statue.

**Summary of evaluation.** The four datasets that we evaluate on include various scene types, ranging from object-level to unbounded scenes, with and without reflections. While some methods perform best on specific datasets, *e.g.*, Geo-NeuS [13] on DTU and Zip-NeRF [4] on Mip-NeRF 360 dataset [3], we show their performance degrades on other types of datasets, *e.g.*, Shiny Blender [44]. In contrast, our method shows competitive performance on all datasets and performs best overall (see averaged metrics in Tab. 2). This demonstrates the robustness of our method to various scene types.

**Custom baselines comparison.** We compare our method with our two custom baselines, CamV and RefV, on the DTU [1], Mip-NeRF 360 [3], and Ref-NeRF real [44] datasets. As shown in Tab. 3, our method outperforms the two baselines in all metrics on all three datasets. Besides, CamV mostly outperforms RefV, while RefV fails on one scene in DTU. This shows that the camera view radiance field is usually more robust than the reflected view radiance field, although this method does not reconstruct the geometry of reflective regions well.

Fig. 5 shows a qualitative comparison, where RefV reconstructs smooth surface for the reflective back window but has artifacts on the side for "sedan", while CamV fails to reconstruct accurate surfaces because of the reflections. On the "toycar" scene, RefV fails to reconstruct the correct geometry, while CamV reconstructs shiny surfaces better while showing artifacts on the hood. For RefV, we sometimes observe optimization issues with separate diffuse and specular components, where the specular component may be blank throughout training and the diffuse component (w./o. directional input) wrongly represents the view-dependent appearance with incorrect geometry (see supp. mat. for details). By coupling two difference radiance fields continuously in 3D, our method represents the appearance and geometry better than the baselines that only use a single radiance field, leading to higher-quality reconstructed surfaces.

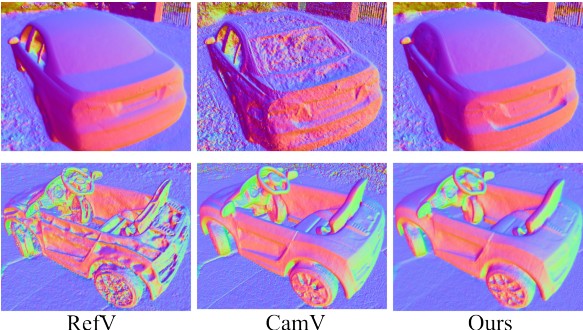

| RefV | CamV | Ours |

Figure 5: Qualitative comparison of surface normals with two baselines, RefV and CamV on "sedan" and "toycar" scenes [44]. Best viewed when zoomed in.

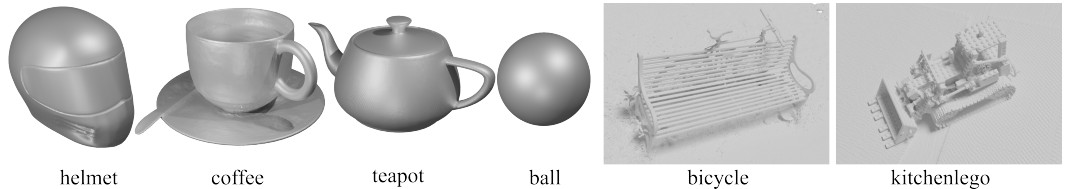

| helmet | coffee | teapot | ball | bicycle | kitchenlego |

Figure 6: Visualization of our meshes. Best viewed when zoomed in.

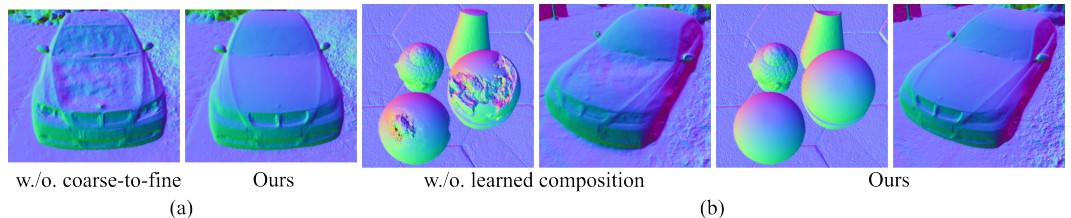

| w./o. coarse-to-fine | Ours | w./o. learned composition | Ours |
| (a) | | (b) | |

Figure 7: Ablation study of our method. Best viewed when zoomed in.

**Mesh visualization.** In Fig. 6, we visualize our meshes on Shiny Blender [44] and Mip-NeRF 360 dataset [3]. Our method can accurately reconstruct the surfaces of reflective objects as well as large-scale scenes with fine geometric details.

### 4.3 Ablation Study

**Coarse-to-fine training.** As shown in Fig. 7 (a), the reconstructions of "sedan" [44] contain artifacts on the specular window and hood without training in a coarse-to-fine manner. With all feature pyramid grids activated in the beginning, the hash grid backbone can easily overfit to the specular effects with wrong geometry.

**Unification of radiance fields.** In Tab. 3, we show that composing two radiance fields can consistently improve performance. In this ablation, we justify the effectiveness of unification with learnable weights. Since the main difference of two radiance fields is the view directional input, we design a baseline with a single radiance field that takes both camera view $\mathbf{d}$ and reflected view $\omega_r$ as inputs. Structurally, this baseline has the same input information as our radiance fields and weight field. As shown in Fig. 7 (b), our method performs better in reconstruction, while the baseline cannot reconstruct reflective surfaces well on both scenes.

## 5 Conclusion

In this paper, we have presented UniSDF, a novel algorithm that learns to seamlessly combine radiance fields for robust and accurate reconstruction of complex scenes with reflections. We find that camera

view radiance fields, *e.g.*, NeRF, are robust to complex real settings but cannot reconstruct reflective surfaces well, while reflected view radiance fields, *e.g.*, Ref-NeRF, can effectively reconstruct highly specular surfaces but struggle in real-world settings and to represent other types of surfaces. By adaptively combining camera view and reflected view radiance fields with a learnable weight field, our method significantly outperforms the baselines with either single radiance field. Together with a hash grid backbone to accelerate training and improve reconstruction details, our method performs superior to or on par with state-of-the-art methods, tailored for handling reflections or not, in reconstruction and rendering on different types of scenes, ranging from object-level to unbounded scenes, with and without reflections.

**Acknowledgement.**   We would like to thank Dor Verbin, Peter Hedman, Ben Mildenhall and Pratul P. Srinivasan for feedback and comments.

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

# A    Appendix / supplemental material

In the supplementary, we first discuss more experimental details, including dataset and implementation details. Second, we discuss BakedSDF [51] in more detail. We show that BakedSDF is often not stable and we discuss how we finetune this method to improve performance on several scenes. Third, we summarize the detailed evaluation results on DTU [1], Shiny Blender [44], Mip-NeRF 360 [3] and Ref-NeRF real [44] datasets. We also qualitatively compare with the state-of-the-art-methods [13, 51, 21, 14, 4]. Fourth, we include more ablation study. Fifth, we discuss the comparison with two custom baselines, RefV and CamV, in more detail. Finally, we discuss the potential societal impacts and limitations of our method.

# B    Experimental Settings

## B.1    Dataset Details

We use the same data splits as prior works for fair comparison. On DTU [1], following [47, 13], we use all the images for surface reconstruction. On Shiny Blender [44], Mip-NeRF 360 [3], and Ref-NeRF real [44] datasets, we follow the official protocol and use the official training / testing split for training and testing.

## B.2    Implementation Details

**Network Architecture.**    In addition to the iNGP [30] structure that we have introduced in the main paper, we further discuss the details of the MLP architectures. Specifically, the SDF MLP $f$ has 2 layers with 256 hidden units and outputs the bottleneck feature vector $\mathbf{b}$ with size 256. The two radiance MLP $f_{cam}$, $f_{ref}$ have 4 layers with 256 hidden units. Besides, the weight MLP $f_w$ has a single layer with 256 hidden units.

Recall that following Mip-NeRF 360 [3], we use two rounds of proposal sampling and then a final NeRF sampling round. The proposal sampling is used to bound the scene geometry and recursively generate more detailed sample intervals, while the final NeRF sampling is used to render the final set of intervals into an image. We set the number of samples for these 3 sampling rounds as 64, 32, 32 for the object-level DTU [1] and Shiny Blender [44], and 64, 64, 32 for unbounded Mip-NeRF 360 [3] and Ref-NeRF real [44] datasets.

In Sec. 3 of the main paper, we mainly introduce model details of the final NeRF sampling round, where the color is rendered, for simplicity. Thus, we introduce the details for proposal sampling rounds here. Specifically, the proposal sampling rounds only have a SDF MLP, *i.e.*no radiance MLP and weight MLP, since color is not rendered in these rounds. Moreover, the two proposal sampling rounds share a SDF MLP, which is different from the SDF MLP in the NeRF sampling round. Contrary to Zip-NeRF [4] that uses a distinct iNGP for each sampling round, we use a single iNGP that is shared by all sampling rounds. We find that this produces similar performance as using multiple iNGPs but explicitly simplifies the model.

**Loss Function.**    In the loss function (Eq. 12 in the main paper, which is for the final NeRF sampling round), we set $\lambda_1 = 10^{-4}$ and $\lambda_3 = 10^{-3}$. Moreover, we set $\lambda_2 = 10^{-4}$ for Shiny Blender [44], and $\lambda_2 = 10^{-3}$ for DTU [1], Mip-NeRF 360 [3] and Ref-NeRF real [44] datasets. For proposal sampling rounds, we replace $\mathcal{L}_{\text{color}}$ with $\mathcal{L}_{\text{prop}}$, the proposal loss described in Mip-NeRF 360 [3].

**Training Details.**    For training, we use the Adam [20] optimizer with $\beta_1 = 0.9, \beta_2 = 0.999, \epsilon = 10^{-6}$. We warm up the learning rate in the first $2\%$ iterations and then decay the it logarithmically from $5 \times 10^{-3}$ to $5 \times 10^{-4}$.

# C    BakedSDF

In our experiments, we find that the optimization of BakedSDF [51] is sensitive and often fails completely on Shiny Blender [44] and Ref-NeRF real dataset [44], as shown in Fig. 8.

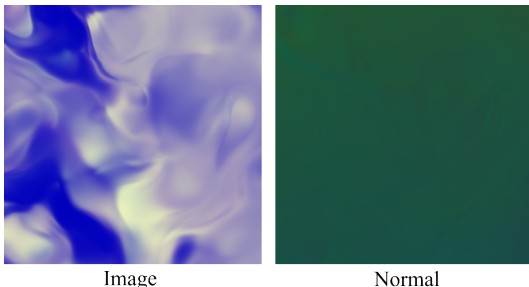

| Image | Normal |

Figure 8: Final image rendering and normal of original BakedSDF [51] on "garden spheres" scene [44]. The training is not stable leading to degraded results (see text).

BakedSDF only uses eikonal loss, $\mathcal{L}_{\text{eik}}$, for regularization, where the corresponding loss weight is set to 0.1 by default. We experimentally find that decreasing the eikonal loss weight can stabilize the training and thus carefully tune it for each scene. For Shiny Blender [44], we set the eikonal loss weight as $10^{-2}$ for "toaster", "helmet" and "coffee", and $10^{-1}$ for "ball". Unfortunately, we could not find the best eikonal loss weight for "car" and "teapot" scenes since the training does not produce reasonable geometry. For completeness, we report the rendering metrics on these two scenes with eikonal loss weight as $10^{-2}$. For Ref-NeRF real dataset [44], we set the eikonal loss weight as $10^{-2}$ for "sedan" and "toycar", and $10^{-5}$ for "garden spheres".

## D  Detailed Evaluation Results

Tab. 4, Tab. 5, Tab. 6 and Tab. 7 contain the detailed metrics for each individual scene on DTU [1], Shiny Blender [44], Mip-NeRF 360 [3] and Ref-NeRF real [44] datasets respectively.

We qualitatively compare with state-of-the-art methods [21, 51, 4] on Shiny Blender, Mip-NeRF 360 and Ref-NeRF real datasets in Fig. 9. Our method is robust and demonstrates competitive performance on various scene types, ranging from object-level to unbounded scenes, with and without reflections.

**Shiny Blender.**  In addition to the MAE error for evaluating the surface normal accuracy, we also evaluate the mesh quality. Though Chamfer Distance includes both *accuracy* and *completeness*, Ref-NeuS [14] points out that the ground-truth meshes of Shiny Blender [44] are double-layered, which results in many redundant points in the ground truth. Therefore, we follow Ref-NeuS [14] and only evaluate the *accuracy* (Acc) of reconstructed meshes. Besides, since ground-truth meshes of "ball" and "teapot" are unavailable, we only evaluate on the remaining four scenes following Ref-NeuS.

As shown in Fig. 10, since the objects in Shiny Blender are highly reflective, our method automatically assigns high weights for the reflected view radiance field in most regions. Though our learned weight is not supervised, it can successfully detect the reflective surfaces. Besides, our reflected view radiance field represents the highly specular reflections of the surrounding environment, which is similar to the observation in Fig. 3 of the main paper.

We further qualitatively compare the surface normals with Geo-NeuS [13], Neuralangelo [21] and Ref-NeuS [14] in Fig. 11. For "teapot" scene, Geo-NeuS fails to reconstruct the smooth surface of the teapot. Neuralangelo reconstructs incorrect surface elements under the object. Ref-NeuS reconstructs overly smooth surfaces without fine details. For "ball" scene, Geo-NeuS produces slight artifacts on the surface, while Neuralangelo reconstructs lots of incorrect surface elements inside the ball. In contrast, our method reconstructs the surfaces more accurately on both scenes.

## E  Custom Baselines Comparison

In the main paper, we have compared our method with two custom baselines, CamV and RefV, both quantitatively (Tab. 3) and qualitatively (Fig. 5).

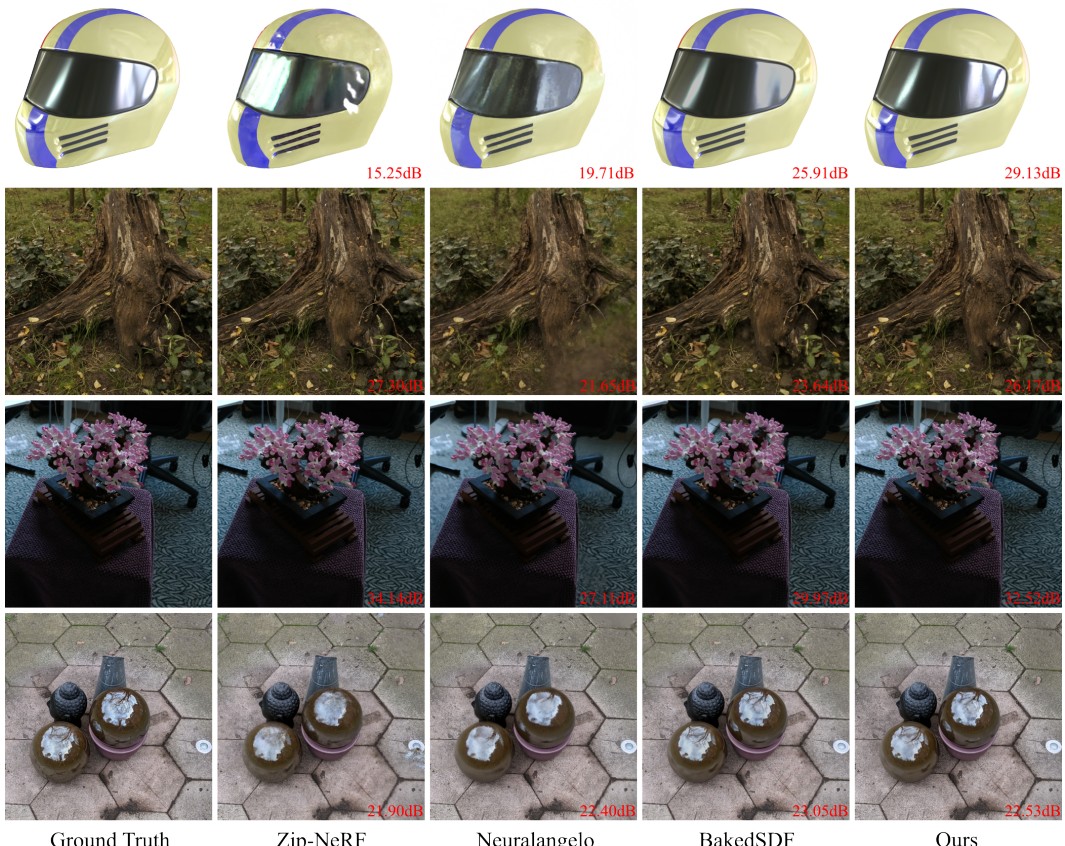

|  |  |  |  |  |
|---|---|---|---|---|
| Ground Truth | Zip-NeRF | Neuralangelo | BakedSDF | Ours |

Figure 9: Qualitative comparison with state-of-the-art methods [21, 51, 4] on Shiny Blender [44], Mip-NeRF 360 [3] and Ref-NeRF real [44] datasets. PSNR values for each image patch are inset. Best viewed when zoomed in.

Table 4: Quantitative Chamfer distance (↓) of individual scenes on DTU dataset [1]. Red, orange and yellow indicate the first, second and third best performing methods for each scene. †: Factored-NeuS [12] provides valid results for 14 scenes, except for scan 69.

| Methods | 24 | 37 | 40 | 55 | 63 | 65 | 69 | 83 | 97 | 105 | 106 | 110 | 114 | 118 | 122 |
|---|---|---|---|---|---|---|---|---|---|---|---|---|---|---|---|
| NeuS [47] | 1.37 | 1.21 | 0.73 | 0.40 | 1.20 | 0.70 | 0.72 | 1.01 | 1.16 | 0.82 | 0.66 | 1.69 | 0.39 | 0.49 | 0.51 |
| NeuralWarp [11] | 0.49 | 0.71 | 0.38 | 0.38 | 0.79 | 0.81 | 0.82 | 1.20 | 1.06 | 0.68 | 0.66 | 0.74 | 0.41 | 0.63 | 0.51 |
| Geo-NeuS [13] | 0.38 | 0.53 | 0.34 | 0.36 | 0.80 | 0.45 | 0.41 | 1.03 | 0.84 | 0.55 | 0.46 | 0.47 | 0.29 | 0.36 | 0.35 |
| Neuralangelo [21] | 0.49 | 1.05 | 0.95 | 0.38 | 1.22 | 1.10 | 2.16 | 1.68 | 1.78 | 0.93 | 0.44 | 1.46 | 0.41 | 1.13 | 0.97 |
| NERO [25] | 1.10 | 1.13 | 1.26 | 0.46 | 1.32 | 1.93 | 0.87 | 1.61 | 1.47 | 1.10 | 0.70 | 1.14 | 0.39 | 0.52 | 0.57 |
| Ref-NeuS [14] | 1.17 | 4.26 | 1.32 | 0.43 | 4.41 | 1.11 | 3.19 | 1.45 | 3.46 | 1.20 | 0.74 | 1.94 | 0.49 | 3.21 | 0.66 |
| Factored-NeuS† [12] | 0.82 | 1.05 | 0.85 | 0.40 | 0.99 | 0.59 | - | 1.44 | 1.15 | 0.81 | 0.58 | 0.89 | 0.36 | 0.44 | 0.46 |
| Ours | 0.54 | 0.84 | 0.66 | 0.51 | 0.76 | 0.64 | 0.71 | 0.70 | 0.86 | 0.57 | 0.69 | 0.65 | 0.45 | 0.56 | 0.50 |

In Fig. 12, we further visualize the qualitative results on scan 37 of DTU [1]. On the one hand, RefV reconstructs holes on the objects with or without reflections, which is similar to the artifacts that BakedSDF [51] shows in Fig. 4 of the main paper. On the other hand, though the surface is a little noisy, CamV reconstructs shiny objects relatively well. Note that in Fig. 5 of the main paper, CamV also reconstructs the shiny surfaces of "toycar" well, despite having some small artifacts. Since scan 37 of DTU and "toycar" of Ref-NeRF real dataset mainly contain reflective surfaces that are less specular, we can infer that camera view radiance field can handle less specular reflections to some extent.

As shown in Fig. 13, we sometimes observe that RefV has optimization issues with separate diffuse and specular components. Specifically, the specular component may be empty throughout training,

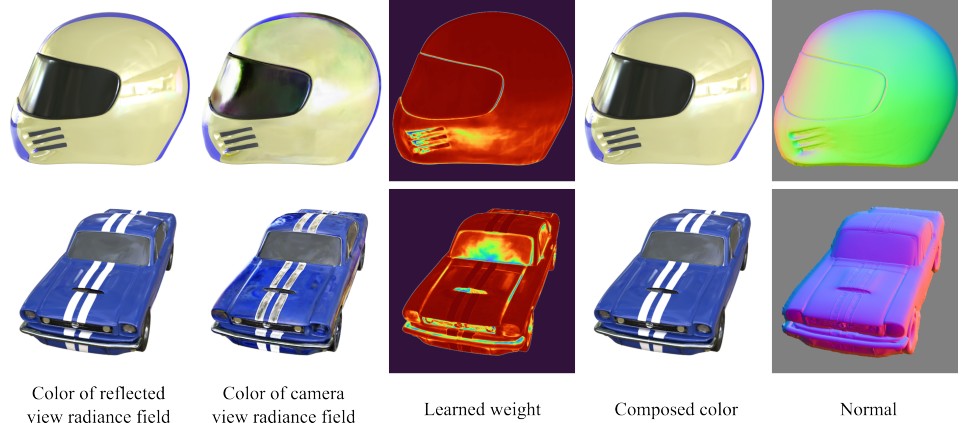

| Color of reflected view radiance field | Color of camera view radiance field | Learned weight | Composed color | Normal |

Figure 10: Visualization of the color of reflected view radiance field, color of camera view radiance field, learned weight **W** (red color represents large weight), composed color and surface normal on "helmet" and "car" scenes [44]. Best viewed when zoomed in.

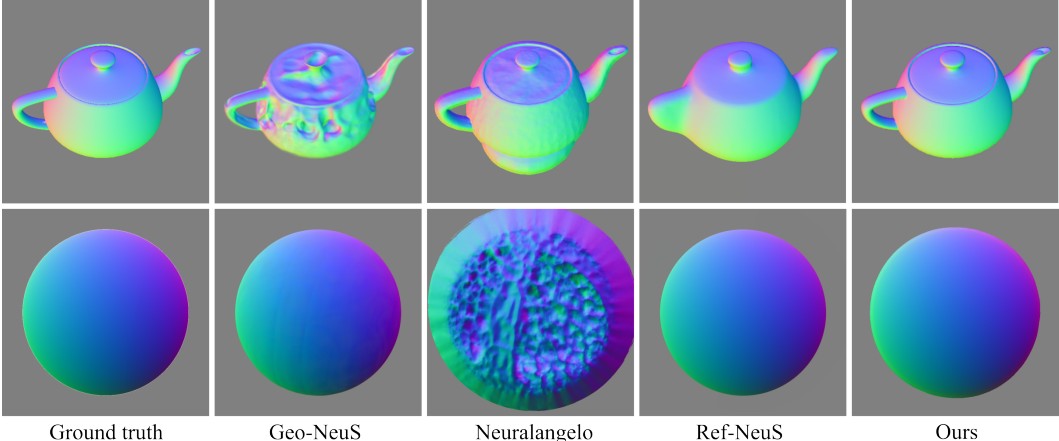

| Ground truth | Geo-NeuS | Neuralangelo | Ref-NeuS | Ours |

Figure 11: Qualitative comparison of surface normal on "teapot" and "ball" scenes [44]. Best viewed when zoomed in.

while the diffuse component represents both view-dependent and non view-dependent appearance. Since diffuse component depends only on the 3D position $\mathbf{x}$, the view-dependence is represented with incorrect geometry, as shown in the results of "toycar" in Fig. 13. We believe that this issue may be related to the high frequency signals that iNGP [30] can encode. In RefV, the diffuse component is parameterized by the feature vector $\gamma$ from iNGP, which is capable of representing very high frequency signal. Therefore, the diffuse component may take advantage of the high capacity of the iNGP representation to model the view-dependent appearance with geometry only, leading to an incorrect reconstruction. This is especially true for small-scale scenes with relatively simple view-dependent appearance, *e.g.*, "toycar".

## F   Ablation Study

**Normal.**   Recall that we use normal $\mathbf{n}$, the gradient of the signed distance field, for computing reflected view direction $\omega_r$ and loss function Eq. 11. In contrast, Ref-NeRF [44] uses predicted normal $\mathbf{n}'$ instead. In this ablation study, we follow Ref-NeRF [44] and use the predicted normal $\mathbf{n}'$ for computing reflected view direction $\omega_r$ and loss function Eq. 11. As shown in Tab. 8, our method performs better than the baseline in rendering. We visualize the surface normal in Fig. 14. Our method successfully reconstructs the reflective surfaces, while the baseline reconstructs artifacts on them.

Table 5: Quantitative results of individual scenes on Shiny Blender [44]. BakedSDF [51] fails on "car" and "teapot" scenes without producing reasonable geometry. Thus we do not report its MAE metric on these scenes. *: We follow Ref-NeuS [14] and evaluate *accuracy* of mesh on four scenes (car, helmet, toaster, coffee). Red, orange, and yellow indicate the first, second, and third best performing algorithms for each scene.

| Methods | car | ball | helmet | teapot | toaster | coffee |
|---|---|---|---|---|---|---|
| **PSNR ↑** | | | | | | |
| Mip-NeRF 360 [3] | 26.48 | 27.69 | 27.41 | 17.84 | 22.47 | 31.07 |
| Zip-NeRF [4] | 27.46 | 21.70 | 26.87 | 45.45 | 23.23 | 30.76 |
| Geo-NeuS [13] | 26.92 | 35.50 | 25.64 | 32.62 | 23.49 | 28.49 |
| Neuralangelo [21] | 27.58 | 27.87 | 28.68 | 44.38 | 23.42 | 32.16 |
| Ref-NeRF [44] | 30.82 | 47.46 | 29.68 | 47.90 | 25.70 | 34.21 |
| ENVIDR [22] | 29.88 | 41.03 | 36.98 | 46.14 | 26.63 | 34.45 |
| NERO [25] | 25.53 | 30.26 | 29.20 | 38.70 | 26.46 | 28.89 |
| Ref-NeuS [14] | 24.30 | 34.57 | 28.07 | 28.73 | 22.68 | 26.09 |
| BakedSDF [51] | 10.03 | 31.35 | 35.50 | 17.84 | 23.84 | 35.06 |
| Ours | 29.86 | 44.10 | 38.84 | 48.76 | 26.18 | 33.17 |
| **SSIM ↑** | | | | | | |
| Mip-NeRF 360 [3] | 0.922 | 0.937 | 0.939 | 0.967 | 0.900 | 0.966 |
| Zip-NeRF [4] | 0.932 | 0.906 | 0.946 | 0.996 | 0.910 | 0.966 |
| Geo-NeuS [13] | 0.922 | 0.982 | 0.946 | 0.985 | 0.884 | 0.951 |
| Neuralangelo [21] | 0.935 | 0.931 | 0.953 | 0.995 | 0.910 | 0.970 |
| Ref-NeRF [44] | 0.955 | 0.995 | 0.958 | 0.998 | 0.922 | 0.974 |
| ENVIDR [22] | 0.972 | 0.997 | 0.993 | 0.999 | 0.955 | 0.984 |
| NERO [25] | 0.949 | 0.974 | 0.971 | 0.995 | 0.929 | 0.956 |
| Ref-NeuS [14] | 0.919 | 0.989 | 0.971 | 0.981 | 0.903 | 0.941 |
| BakedSDF [51] | 0.807 | 0.979 | 0.990 | 0.967 | 0.939 | 0.978 |
| Ours | 0.954 | 0.993 | 0.990 | 0.998 | 0.945 | 0.973 |
| **LPIPS ↓** | | | | | | |
| Mip-NeRF 360 [3] | 0.062 | 0.189 | 0.127 | 0.094 | 0.144 | 0.118 |
| Zip-NeRF [4] | 0.060 | 0.231 | 0.115 | 0.010 | 0.127 | 0.128 |
| Geo-NeuS [13] | 0.080 | 0.082 | 0.082 | 0.024 | 0.134 | 0.110 |
| Neuralangelo [21] | 0.066 | 0.186 | 0.085 | 0.017 | 0.123 | 0.091 |
| Ref-NeRF [44] | 0.041 | 0.059 | 0.075 | 0.004 | 0.095 | 0.078 |
| ENVIDR [22] | 0.031 | 0.020 | 0.022 | 0.003 | 0.097 | 0.044 |
| NERO [25] | 0.074 | 0.094 | 0.050 | 0.012 | 0.089 | 0.110 |
| Ref-NeuS [14] | 0.076 | 0.058 | 0.046 | 0.025 | 0.114 | 0.119 |
| BakedSDF [51] | 0.197 | 0.094 | 0.019 | 0.079 | 0.079 | 0.072 |
| Ours | 0.047 | 0.039 | 0.021 | 0.004 | 0.072 | 0.078 |
| **MAE° ↓** | | | | | | |
| Geo-NeuS [13] | 12.15 | 1.04 | 4.12 | 19.77 | 16.23 | 9.79 |
| Neuralangelo [21] | 12.34 | 35.63 | 9.23 | 4.98 | 14.14 | 8.61 |
| Ref-NeRF [44] | 14.93 | 1.55 | 29.48 | 9.23 | 42.87 | 12.24 |
| BakedSDF [51] | - | 0.44 | 1.74 | - | 12.24 | 3.31 |
| ENVIDR [22] | 7.10 | 0.74 | 1.66 | 2.47 | 6.45 | 9.23 |
| Ref-NeuS [14] | 7.68 | 0.50 | 1.94 | 10.25 | 5.95 | 5.73 |
| Ours | 6.88 | 0.45 | 1.72 | 2.80 | 8.71 | 8.00 |
| **Acc* ↓** | | | | | | |
| Geo-NeuS [13] | 0.72 | - | 0.74 | - | 4.14 | 0.90 |
| Neuralangelo [21] | 1.89 | - | 1.71 | - | 2.99 | 0.63 |
| Ref-NeuS [14] | 0.50 | - | 0.53 | - | 0.54 | 1.83 |
| Ours | 0.58 | - | 0.46 | - | 2.02 | 1.17 |

# G   Potential Societal Impacts

**Positive impact.**   Our method can accurately reconstruct complex scenes with reflections, which is a challenging task for existing methods. The accurate reconstruction can be used for many downstream applications, *e.g.*, robotics and creating 3D experiences for augmented/virtual reality.

**Negative impact.**   Similar to most existing volumetric implicit methods [28, 3, 4, 21, 44], our method needs to be individually trained for each scene. Though the training is fast (3.50h on Mip-

Table 6: Quantitative results of individual scenes on Mip-NeRF 360 dataset [3]. Red, orange, and yellow indicate the first, second, and third best performing algorithms for each scene.

| **PSNR** | Outdoor | | | | | Indoor | | | |
|---|---|---|---|---|---|---|---|---|---|
| | bicycle | flowers | garden | stump | treehill | room | counter | kitchen | bonsai |
| Mip-NeRF 360 [3] | 24.40 | 21.64 | 26.94 | 26.36 | 22.81 | 31.40 | 29.44 | 32.02 | 33.11 |
| Zip-NeRF [4] | 25.80 | 22.40 | 28.20 | 27.55 | 23.89 | 32.65 | 29.38 | 32.50 | 34.46 |
| Neuralangelo [21] | 23.78 | 21.03 | 23.76 | 21.38 | 22.83 | 28.76 | 25.39 | 30.40 | 28.39 |
| BakedSDF [51] | 23.05 | 20.55 | 26.44 | 24.39 | 22.55 | 30.68 | 27.99 | 30.91 | 31.26 |
| Ours | 24.67 | 21.83 | 27.46 | 26.39 | 23.51 | 31.25 | 29.26 | 31.73 | 32.86 |

| **SSIM** | Outdoor | | | | | Indoor | | | |
|---|---|---|---|---|---|---|---|---|---|
| | bicycle | flowers | garden | stump | treehill | room | counter | kitchen | bonsai |
| Mip-NeRF 360 [3] | 0.693 | 0.583 | 0.816 | 0.746 | 0.632 | 0.913 | 0.895 | 0.920 | 0.939 |
| Zip-NeRF [4] | 0.769 | 0.642 | 0.860 | 0.800 | 0.681 | 0.925 | 0.902 | 0.928 | 0.949 |
| Neuralangelo [21] | 0.605 | 0.508 | 0.635 | 0.502 | 0.623 | 0.869 | 0.796 | 0.891 | 0.857 |
| BakedSDF [51] | 0.588 | 0.504 | 0.793 | 0.662 | 0.543 | 0.892 | 0.845 | 0.903 | 0.911 |
| Ours | 0.737 | 0.606 | 0.844 | 0.759 | 0.670 | 0.914 | 0.888 | 0.919 | 0.939 |

| **LPIPS** | Outdoor | | | | | Indoor | | | |
|---|---|---|---|---|---|---|---|---|---|
| | bicycle | flowers | garden | stump | treehill | room | counter | kitchen | bonsai |
| Mip-NeRF 360 [3] | 0.289 | 0.345 | 0.164 | 0.254 | 0.338 | 0.211 | 0.203 | 0.126 | 0.177 |
| Zip-NeRF [4] | 0.208 | 0.273 | 0.118 | 0.193 | 0.242 | 0.196 | 0.185 | 0.116 | 0.173 |
| Neuralangelo [21] | 0.390 | 0.419 | 0.326 | 0.440 | 0.360 | 0.269 | 0.310 | 0.172 | 0.312 |
| BakedSDF [51] | 0.400 | 0.437 | 0.204 | 0.343 | 0.471 | 0.270 | 0.293 | 0.165 | 0.244 |
| Ours | 0.243 | 0.320 | 0.136 | 0.242 | 0.265 | 0.206 | 0.206 | 0.124 | 0.184 |

Table 7: Quantitative results of individual scenes on the Ref-NeRF real dataset [44]. Red, orange, and yellow indicate the first, second, and third best methods for each metric.

| Methods | Sedan | | | Toycar | | | Garden Spheres | | |
|---|---|---|---|---|---|---|---|---|---|
| | PSNR ↑ | SSIM ↑ | LPIPS ↓ | PSNR ↑ | SSIM ↑ | LPIPS ↓ | PSNR ↑ | SSIM ↑ | LPIPS ↓ |
| Mip-NeRF 360 [3] | 25.56 | 0.708 | 0.304 | 24.32 | 0.654 | 0.256 | 22.94 | 0.587 | 0.268 |
| Zip-NeRF [4] | 25.85 | 0.733 | 0.260 | 23.41 | 0.626 | 0.243 | 21.77 | 0.545 | 0.238 |
| Neuralangelo [21] | 24.82 | 0.656 | 0.384 | 24.28 | 0.638 | 0.293 | 22.03 | 0.529 | 0.313 |
| Ref-NeRF [44] | 25.20 | 0.639 | 0.406 | 24.40 | 0.627 | 0.292 | 22.57 | 0.502 | 0.366 |
| BakedSDF [51] | 25.70 | 0.700 | 0.332 | 24.51 | 0.655 | 0.280 | 23.08 | 0.553 | 0.363 |
| Ours | 24.68 | 0.700 | 0.309 | 24.15 | 0.639 | 0.245 | 22.27 | 0.567 | 0.243 |

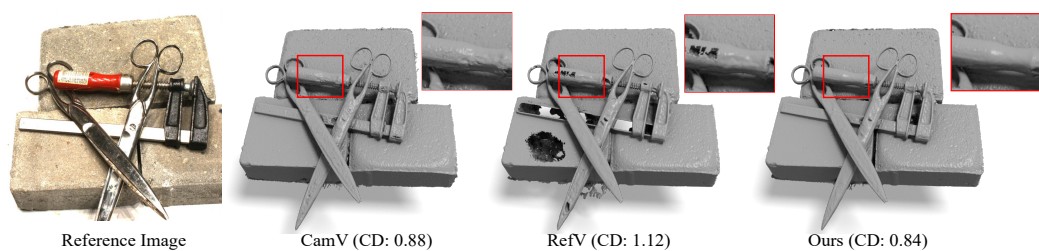

| Reference Image | CamV (CD: 0.88) | RefV (CD: 1.12) | Ours (CD: 0.84) |
|---|---|---|---|

Figure 12: Comparison with two baselines, CamV and RefV, on scan 37 of DTU [1] (CD is Chamfer distance error). CamV reconstructs more noisy surface on the red handle with reflections (highlighted with red box and zoomed in), while RefV generates holes on the shiny objects and even the brick without any reflections. Best viewed when zoomed in.

NeRF 360 [3] and Ref-NeRF real [44] datasets), we use 8 NVIDIA Tesla V100-SXM2-16GB GPUs simultaneously, which consumes much energy.

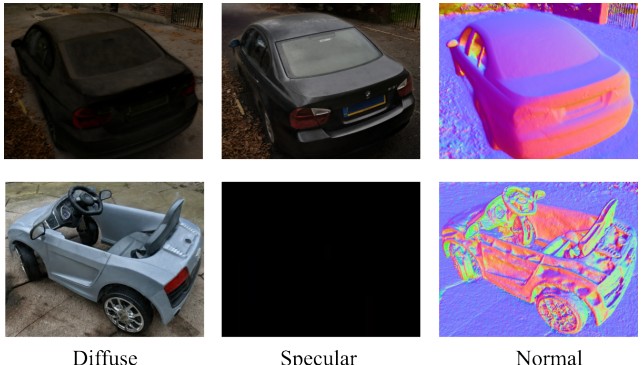

| Diffuse | Specular | Normal |

Figure 13: Visualization of diffuse color component, specular color component and surface normal for RefV on "sedan" and "toycar" scene [44]. RefV successfully decomposes two color components for "sedan", while it fails on "toycar" with blank specular component. Best viewed when zoomed in.

Table 8: Ablation study of normals on the Ref-NeRF real dataset [44].

| Methods | PSNR ↑ | SSIM ↑ | LPIPS ↓ |
|---|---|---|---|
| w. predicted normal | 23.56 | 0.630 | 0.268 |
| Ours | 23.70 | 0.636 | 0.265 |

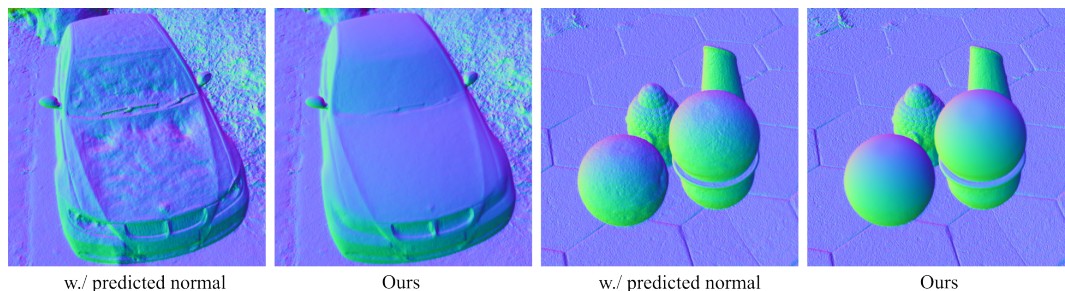

| w./ predicted normal | Ours | w./ predicted normal | Ours |

Figure 14: Ablation study of normals on "sedan" and "garden spheres" scene [44]. Best viewed when zoomed in.

## H  Limitations

In this work, we mainly focus on robust surface reconstruction of scenes with reflections. Our method cannot be used for editing tasks such as relighting. Besides, similar to most NeRF methods [3, 21, 4], our method requires the estimated poses from SfM, *e.g*., COLMAP [38], as input for real-world scenes. However, SfM needs to perform feature matching, which is based on multi-view photometric consistency, for pose estimation. As we discuss in the main paper, multi-view photometric consistency is not guaranteed with reflective surfaces and thus SfM may produce inaccurate poses, which may lead to inaccurate reconstruction and rendering. In addition, our method cannot accurately reconstruct the reflective surfaces with sparse input views. It is challenging to determine reflections in this case because of the ambiguity. Note that both Shiny Blender [44] and Ref-NeRF real dataset [44] carefully provide dense 360 degree views around the reflective objects so that it is easier to detect the view-dependent reflective appearance.

## I  Licenses for Existing Assets

code:

- multinerf [29]: Apache License 2.0.

datasets:

- DTU [1]: we do not find the license.
- Shiny Blender [44]: coffee (CC-0 license), toaster (CC-BY license), car (CC-0 license), helmet (CC-0 license).
- Mip-NeRF 360 dataset [3]: CC-BY license.
- Ref-NeRF real dataset [44]: it is captured by the authors of Ref-NeRF [44], we do not find the license.

