# OpenReview forum: "UniSDF: Unifying Neural Representations for High-Fidelity 3D Reconstruction of Complex Scenes with Reflections"
_NeurIPS.cc/2024/Conference — NeurIPS 2024 poster_

### Official Review · Reviewer_5Tqb · 2024-06-29

**Soundness:** 3
**Presentation:** 2
**Contribution:** 2
**Rating:** 6
**Confidence:** 4

**Summary:**

This paper proposes UniSDF, an improved NeuS architecture capable of reconstructing photorealistic reflective surfaces and robustly working in real-world scenes. The method models reflective color and base color as separate MLPs, using learned weights to blend them and obtain the final colors. Qualitative and quantitative results on several datasets demonstrate that the method surpasses other baselines and results in fewer geometric artifacts.

**Strengths:**

The method proposes a new network architecture to better represent the reflective radiance field, although most of the concepts are derived from previous works. There are some original contributions, such as the blending between two fields inspired by NeRFRen, which is extended to non-planar surfaces, and the coarse-to-fine training from Neuralangelo, used here to reduce the ambiguity of reflections. Therefore, while the method is novel, its significance is not substantial.

Another strength is the robust and high-quality reconstruction demonstrated in the paper and supplementary material, which surpasses other baselines. This is valuable since this problem is crucial in real-world 3D reconstruction scenarios.

**Weaknesses:**

The biggest weakness is the lack of physical interpretation. In Eq. 8, the final color is a linear combination of the camera view and reflected view radiance fields, resulting in unclear physical meaning for each color component. Although the method focuses on novel view synthesis and geometry reconstruction regardless of the underlying physical components, this lack of physical meaning leads to ambiguity about the assumptions the authors have for the reconstructed scene.

For example, the model doesn't account for how reflections change with different surface roughness, raising the question of whether it will perform well on surfaces with spatially varying roughness. Additionally, the model does not explicitly trace the reflective ray but instead models the color of the reflective ray through an MLP. This raises concerns about its ability to handle near-field lighting conditions effectively.

Furthermore, I am curious about the differences between Ref-NeRF's equation c = diffuse + specular × tint and Eq. 8 c=(1−w) c_cam+w c_ref. Will these two modeling approaches yield similar performance?

**Questions:**

1. Title: It's unclear from the abstract and introduction where the "unifying" concept comes from. "Unifying scene representation" suggests combining several scene representations, but theoretically, it is still a NeuS-based representation.
	2. Missing Citations:
		a. MS-NeRF: Multi-space Neural Radiance Field: This paper also presents a similar idea of linearly combining different color fields.
		b. Several Concurrent Works Regarding Reflective NeRF:
			i. NeRF-Casting: Improved View-Dependent Appearance with Consistent Reflections
			ii. SpecNeRF: Gaussian Directional Encoding for Specular Reflections
		c. Reflection Handling in Gaussian Splatting:
			i. MirrorGaussian: Reflecting 3D Gaussians for Reconstructing Mirror Reflections
			ii. Spec-Gaussian: Anisotropic View-Dependent Appearance for 3D Gaussian Splatting
	3. Illustration of Existing Works: In line 131, I wouldn't call Ref-NeRF[43] as using a single radiance field, as it uses two MLPs to model diffuse color and specular color separately (though the specular MLP is conditioned on the feature of the diffuse MLP)
	4. Hyperparameters in Final Loss: The final loss contains several hyperparameters for controlling the strength of each term. In my experience, these weights play an important role. It would be beneficial to conduct small ablations or illustrate how these weights were decided.
	5. Input Redundancy: The f_ref and f_cam functions take in both the spatial coordinate x and the bottleneck feature b from the SDF MLP. Given that b is already a spatial feature vector, what's the point of adding x as one of the inputs? This seems a bit redundant.
	6. Normals and Orientation Loss: There are two normals being produced: n is the normal from the SDF field, and n′ is the MLP-predicted normal. For Eq. 11, Ref-NeRF uses n′ for orientation loss. It's unclear if this is a typo or if n is actually used for the orientation loss.
	7. Volumetric Rendering SDF: In line 124, it would be better to cite which volumetric rendering SDF system is used. I assume it's NeuS.

**Limitations:**

There are no potential negative society impact.

---

> ### Author Rebuttal · Authors · 2024-08-06
>
> >**Explicitly tracing the reflective ray**
>
> In this work, similar to many recent methods that are tailored to handle reflections [12,14,22,50], we follow Ref-NeRF and parameterize part of view-dependent appearance as a function of the reflected view direction $\mathbf{\omega}\_r$. Explicitly tracing the reflective ray is an interesting direction for us and it is shown effective in the concurrent work NeRF-Casting.
>
> >**Ref-NeRF's model v.s. our model**
>
> Our custom baseline “RefV” uses the same physics based model as Ref-NeRF (L215), where we separate the color into diffuse and specular components. As shown in Table.3, our method outperforms RefV in both surface reconstruction and rendering on 3 datasets. Fig.5 shows that RefV reconstructs artifacts on the surfaces, while our method performs better. We also observe that when using the hash grid backbone, RefV may have optimization issues with separate diffuse and specular components as visualized in Fig.13. We discuss this in L533-543.
>
> >**Q1: Unifying neural representations**
>
> What we mean is that we combine the two neural representations for the radiance field, i.e., camera view and reflected view radiance field. We will rephrase the term “scene representation” in the abstract.
>
> >**Q2: Missing citations**
>
> Thank you for pointing this out. We will add them in our paper. MS-NeRF focuses on view synthesis and combines $K \geq 2$ camera view radiance fields, where each camera view radiance field has a volume density field. It is unclear how to extract the surface since there are $K$ different volume density fields and iso-surfaces can be extracted from each of them. In contrast, we use a single SDF field to represent geometry and directly extract iso-surface from it.
>
> >**Q3: Ref-NeRF as a single radiance field**
>
> Thank you for pointing this out. We will rephrase.
>
> >**Q4: Ablation study on loss weights**
>
> We perform an ablation study on the eikonal loss weight $\lambda\_1$ on Ref-NeRF real dataset and summarize the results as follows. Setting $\lambda\_1=10^{-4}$ produces the best performance. If $\lambda\_1$ is too large (e.g., $\lambda\_1=10^{-2}$), we observe that the training becomes unstable and may fail. This is similar to our findings on BakedSDF (Sec. C).
>
> As shown in [46,49], the normals, estimated as the gradients, of the SDF field are well-defined and more accurate than those computed from the volume density field. Recent BakedSDF [50] and Ref-NeuS [14] combined SDF field with reflected view radiance field to reconstruct reflective surfaces. They found that using eikonal loss $\mathcal{L}\_{\text{eik}}$ for regularization is enough to get accurate normals and thus it is not necessary to use the normal smoothness loss $\mathcal{L}\_p$ or the orientational loss $\mathcal{L}\_o$ like Ref-NeRF. In our method, we still use $\mathcal{L}\_p$ and $\mathcal{L}\_o$ for additional regularization. However, since we use eikonal loss already, we simply set  $\lambda\_2, \lambda\_3$ to relatively small values and do not tune them a lot. We only slightly increase the weight $\lambda\_2$ for normal smoothness loss $\mathcal{L}\_p$ on real data because of the potential noise (L481), similar to Ref-NeRF.
>
> Methods | PSNR $\uparrow$ | SSIM$\uparrow$ | LPIPS$\downarrow$
> --|--|--|--
> $\lambda\_1=10^{-2}$ | 20.99 | 0.509 | 0.413
> $\lambda\_1=10^{-3}$ | 23.24 | 0.611 | 0.295
> $\lambda\_1=10^{-4}$ (Ours) | **23.70** | **0.636** | **0.265**
> $\lambda\_1=10^{-5}$ | 23.62 | 0.633 | 0.268
>
>
> >**Q5: Input redundancy**
>
> It is correct that $\mathbf{b}$ is dependent on $\mathbf{x}$. The reason that we use both $\mathbf{b}$ and $\mathbf{x}$ is to align with the state-of-the-art implicit reconstruction pipelines such as VolSDF [49] and NeuS [46], which use both $\mathbf{b}$ and $\mathbf{x}$ to compute radiance.
>
> >**Q6: Normals and orientation loss**
>
> Since the normal estimated from the volume density gradient are often extremely noisy, Ref-NeRF uses the predicted normal $n’$ throughout its pipeline, e.g., computing reflected view direction $\mathbf{\omega}\_r$, including for orientation loss. As shown in [46,49], the normals, estimated as the gradients, of the SDF field are well-defined and much smoother than those computed from the volume density field. Compared with the noisy normal of the volume density field of Ref-NeRF, we use a SDF field as our geometry representation and thus the normal is already smoother and more accurate. So we use $n$ in our pipeline (also for orientation loss) and only use the predicted normal $n’$ for regularization in Eq.10. Note that BakedSDF [50] and Ref-NeuS [14] also use the SDF normal $n$ for their reflected view radiance fields.
>
> >**Q7: Volumetric rendering the SDF**
>
> Sorry, we missed the reference here. We use VolSDF [49] as our representation (L199).

---

### Official Review · Reviewer_2MRi · 2024-07-11

**Soundness:** 3
**Presentation:** 4
**Contribution:** 2
**Rating:** 6
**Confidence:** 3

**Summary:**

The paper tackles the problem of 3D reconstruction in the presence of highly reflective objects. To address the problem, they propose to learn an SDF-based neural representation. Different from prior work, they use two radiance branches in their representation, one conditioned on the camera viewing direction rotated about the normal and one conditioned on the regular viewing direction. The outputs of the two networks are combined using a learned weight to arrive at the final rendered radiance.

The authors argue that while the reflected direction is better at capturing the details in reflective surfaces, the viewing direction is more robust to real data.

Through extensive experiments, they show their method is competitive with prior art in both the tasks of NVS and mesh recovery (as measured by Chamfer Distance).

**Strengths:**

I like the problem being attempted in the paper, I also like the insight that the viewing direction conditioning is more robust, while the reflected direction produces better geometry for reflective surfaces in controlled environments. The results are also impressive.

Quality: The ablation studies are well/extensively done.

Clarity: The paper is very well written and easy to follow. The figures are all well understandable and the work should be easily reproducible from just the description.

**Weaknesses:**

Significance: I think the method is generally quite a straightforward extension of existing work and the paper's contributions are fairly limited. As I see it both the radiance branches have previously existed and the Ref-NeRF parametrization could be considered more “physically based”, the authors forgo a more mathematically solid setup in favour of an increase in performance.

I think the paper would benefit a bit from providing more insight into the reasons for some of the phenomena i.e. why do the weights separate like that for more specular/diffuse regions even though both are conditioned on some sort of parametrization of the viewing direction etc.

I am sympathetic however to the authors that some of these phenomena just occur and might be hard to reason about, but some of the simulated experiments could provide more insight into how the optimization behaves.

**Questions:**

1) Is there a stopgrad on any components of the normal smoothness loss (predicted vs computed normal in Eq 10)?

2) I see there is an ablation on this, but why do the authors think the weight to mix the outputs of the two radiance branches is necessary, why doesn’t this factor just get automatically learned by the two networks?

3) The authors seem to have ran ENVIDR on the Ref-NeRF real dataset (as evidenced by Fig 1?), but do not include the results in Table 2? Why?

4) Have the authors tried passing in $n^\prime$ instead of $n$ to the radiance network or also calculating $w_r$ about it? If I understand correctly $n^\prime$ is supposed to be “smoother”, perhaps this would make results more robust for just RefV?

**Limitations:**

Limitations have been adequately discussed in the supplement.

---

> ### Author Rebuttal · Authors · 2024-08-06
>
> >**Why learned weights separate specular/diffuse regions without explicit supervision**
>
> In Ref-NeRF, the ablation study shows that when rendering reflective surfaces, using the reflected view direction as the MLP’s input is explicitly better than using the camera view direction of NeRF. In our method, the main difference between two radiance fields is the view directional input. The reflected view radiance field uses reflected view direction as Ref-NeRF, while the camera view radiance field uses camera view direction. Similar to the findings in Ref-NeRF, our reflected view radiance field can model the appearance of reflective regions better than the camera view radiance field. Thus the weight $\mathbf{W}$, which corresponds to the rendered color $\mathbf{C}\_{ref}$ of reflected view radiance field (Eq.8), for these reflective regions increases during optimization to reduce the composed color loss $\mathcal{L}\_{\text{color}}$. Conversely, when the color computed from the camera view radiance field is better, the weight $\mathbf{W}$ decreases.
>
> >**Q1: Stopgrad on any components of the normal smoothness loss**
>
> As in Ref-NeRF, we do not use stopgrad on any components.
>
> >**Q2: Weight field to mix the outputs of the two radiance branches**
>
> Thank you for the suggestion. For this rebuttal, we evaluate our method while removing the weight field. The final color is composed as $\mathbf{C} = \mathbf{C}\_{ref} + \mathbf{C}\_{cam}$. We evaluate on the Ref-NeRF real dataset and summarize the results as follows. We observe that our method performs better than this ablation.
>
> As shown in Fig.3, the learned weight field has the advantage of leading to better interpretability as it can be used to detect highly reflective regions from different viewpoints with volume rendering. This is a challenging task because reflection is an attribute of surfaces that objects or parts of objects have. Therefore, it is difficult to detect such regions using existing semantic models. For example, we tried state-of-the-art open vocabulary semantic segmentation methods with “reflective objects” or “reflective surfaces” as prompt, and they were unable to successfully segment the reflective objects in the image.
>
> Methods | PSNR $\uparrow$ | SSIM$\uparrow$ | LPIPS$\downarrow$
> --|--|--|--
> w./o. weight field | 23.39 | 0.625 | 0.279
> ours | **23.70** | **0.636** | **0.265**
>
>
> >**Q3: Results of ENVIDR**
>
> The metrics and rendered images of ENVIDR are the official results provided by the authors of ENVIDR. On the Ref-NeRF real dataset, ENVIDR is evaluated on “gardenspheres” only. In the limitation section of the ENVIDR paper, it is said that ENVIDR is unable to handle unbounded scenes, while the “sedan” scene of Ref-NeRF real dataset is unbounded.
>
> >**Q4: Use $n’$ instead of $n$**
>
> As shown in [46,49], the normals, estimated as the gradients, of the SDF field are well-defined and much smoother than those computed from the volume density field of NeRF. Compared with the noisy normal of the volume density field of Ref-NeRF, we use a SDF field as our geometry representation and thus the normal is already smoother and more accurate. So we only use $n’$ for regularization in Eq.10. Note that BakedSDF [50] and Ref-NeuS [14] also use the SDF normal $n$ for their reflected view radiance fields.

---

> ### Comment · Reviewer_2MRi · 2024-08-11
>
> Thank you very much for the response to my questions. I especially appreciate the response in the global rebuttal.
>
> I'm not fully satisfied with the answer to the question about specular/diffuse separation--I feel like the authors are just reiterating what's happening, as opposed to providing any reasoning.
>
> I also think since reviewer 5Tqb and I both had questions about this, it would be useful to run a quick (even on a smaller subset of scenes) the effect of using different normals in the reflected view direction and/or loss formulations for the camera ready version.
>
> Nonetheless, I think the experimental results, alongside some of the arguments laid out in the global rebuttal, such as the comment of Ref-NeRF not being a complete physical model (not explicitly handling interreflections, etc), make a stronger case for the paper. I am happy to increase my score.

---

> ### Author Response · Authors · 2024-08-12
>
> Thank you for the comments. We will add an ablation study about the normal that we use, i.e. SDF normal $n$ or predicted normal $n’$, for the reflected view direction and loss formulations in the paper.

---

### Official Review · Reviewer_dks9 · 2024-07-12

**Soundness:** 3
**Presentation:** 3
**Contribution:** 3
**Rating:** 6
**Confidence:** 4

**Summary:**

The paper proposes a method to reconstruct scenes containing both reflective surfaces and non-reflective surfaces with high fidelity. Specifically, it trains a camera view radiance field and a reflected view radiance field separately, combining them by a learnable weight. The method is evaluated on four datasets, which covers different situations including objects and 360 scenes, shiny objects, real-world captures. And the method outperforms or is on par with SOTA on all these datasets, demonstrating its effectiveness.

**Strengths:**

1. Extensive experiments. The method was compared to several SOTA methods and outperforms or is on par with them, showing its ability to reconstruct complicated scenes with reflective surfaces on four datasets.
2. The paper is well written. And the figures clearly serve for their purposes.
3. The author did a great job in analyzing the limitations and social impact, which is highly encouraged.

**Weaknesses:**

1. The method involves two radiance fields and one weight field, and it has to perform volume rendering for colors from both radiance fields and the weights, which is very time-consuming even with implementation of iNGP.
2. I wonder if you have a possible explanation for why some reflected view radiance field methods is not robust to real-world scene.
3. I noticed the comparison with Factored-NeuS is only performed on DTU. Since Factored-NeuS is designed to also handle glossy objects, I wonder if how it look like when compared to this paper on Shiny Blender and Ref-NeRF real datasets.
4. As the authors mention in the section of limitations, this method requires posed images and SfM can fail on reflective surfaces. While this is a common issue, I still wonder how robust this method is to inaccurate camera pose. Even a failure case is fine.

**Questions:**

Please see weaknesses.

**Limitations:**

The authors listed a quite complete list of limitations. I think some are common issues for many works like requiring camera pose and dense views. With that being said, in the future the authors can consider extending this work to sparse views and how to eliminate the requirement for several fields and volume rendering, making the algorithm more efficient.

---

> ### Author Rebuttal · Authors · 2024-08-06
>
> >**W1: Time consuming**
>
> Our method has three fields and is based on volume rendering. Thus our rendering efficiency is not high. Recently, 3D Gaussian Splatting has become popular in view synthesis and there are some concurrent works focusing on reconstructing the surface [1*] or rendering reflections [2*]. Therefore, we think extending our pipeline with 3D Gaussian Splatting is a potential future direction to improve efficiency. Thank you for your suggestion to explore eliminating the multiple fields, this is an interesting direction for future works.
>
> >**W2: Explanation for why some reflected view radiance field methods are not robust to real-world scene**
>
> Disambiguating the influence of geometry, color and reflection is an ill-posed problem in image-based 3D reconstruction. Methods such as Ref-NeRF assume that there are no inter-reflections or non-distant illumination, which is not often the case in real world data. Please see the global rebuttal for more details.
>
> For the reflected view radiance field, the view-dependent appearance mainly depends on $\mathbf{\omega}_r$, the reflected view direction around the normal $\mathbf{n}$. However, the geometry and surface normal $\mathbf{n}$ are unknown in the beginning and need to be optimized during training. Therefore, the directional input $\mathbf{\omega}_r$ of view-dependent appearance keeps changing during training. We conjecture that this makes the optimization process more complex and ill-posed.
>
> >**W3: Results of Factored-NeuS**
>
> The metrics of Factored-NeuS on DTU are the official results from their paper (the public ICLR 2024 submission). We just found that Factored-NeuS was open-sourced recently. Thus we evaluate Factored-NeuS on ShinyBlender dataset and summarize the results as follows. Our method performs better than Factored-NeuS in both rendering and surface accuracy. As shown in Fig.2 of the rebuttal PDF, Factored-NeuS reconstructs explicit artifacts on the “helmet” scene, while our method reconstructs the surface more accurately.
>
> Methods  |  PSNR $\uparrow$  |  SSIM$\uparrow$  |  LPIPS$\downarrow$  | MAE $\downarrow$  |  Acc $\downarrow$
> --|--|--|--|--|--
> Factored-NeuS  | 30.89 |  0.954 |  0.076 |  5.31 | 1.90
> Ours | **36.82**  |  **0.976**  |  **0.043** |  **4.76**  | **1.06**
>
>
> >**W4: Results with inaccurate camera pose**
>
> Thank you for the suggestion. In this rebuttal, we evaluate some scenes of the Ref-NeRF real dataset on camera poses with uniform noise.  Specifically, we first add random translation noise (with a range as [-0.01,0.01]) along each axis for each camera pose. Note that before adding noise, the scene is already normalized to a unit cube following Mip-NeRF360. Second, we add random rotation noise to the camera poses. For each camera pose, we randomly sample a 3d unit vector and rotate the camera pose around it  (with a range as [-1,1] degrees). As shown in Fig.3 of the rebuttal PDF, both the rendering and surface become worse. Integrating existing methods for pose optimization during training could be an interesting future direction.
>
> [1*] Huang et al. 2D Gaussian Splatting for Geometrically Accurate Radiance Fields.  SIGGRAPH, 2024
>
> [2*] Jiang et al. GaussianShader: 3D Gaussian Splatting with Shading Functions for Reflective Surfaces. CVPR 2024

---

> > ### Comment · Reviewer_dks9 · 2024-08-14
> >
> > I appreciate the author's efforts on answering my questions especially providing additional results in such a short time. My concerns / questions are solved. I'll keep my positive rating.

---

### Official Review · Reviewer_aZTE · 2024-07-17

**Soundness:** 4
**Presentation:** 4
**Contribution:** 3
**Rating:** 7
**Confidence:** 4

**Summary:**

This paper proposes a new strategy for modelling view-dependent effects in Neural Radiance Field-based scene models. Existing approaches have used networks conditioned on camera view directions, as well as on reflected view directions using surface normals, but this work proposes and validates the idea that both model types should be used simultaneously. This is achieved by employing a learned weight model that mixes the two color predictions, thereby allowing regions where the color is not well explained by the reflection-based model to be handled by the view direction-based model. This is shown through ablations to be a better strategy than either strategy individually, and is also compared to a number of baseline methods which employ reflection-based modelling of view-dependent effects.

**Strengths:**

This paper presents a simple, yet effective idea that would be easy to incorporate into other works which use reflection-based models. I think a lot of the contribution is in showing such a simple but surprising result which is not obvious to try.

The quality and clarity of the writeup is high, and the evaluation seems quite thorough in comparing to other relevant works and on relevant datasets. I think the value of the proposed strategy is shown quite clearly by the experiments.

**Weaknesses:**

The only notable weakness is that the proposed strategy is a fairly minor deviation from previously proposed methods. However, I think this is largely mitigated by the non-obvious nature of the change and the in-depth experimental evaluation.

**Questions:**

Given the increasing prevalence of methods like 3DGS which do not use neural networks for modelling view-dependence but rather simpler approaches like spherical harmonics, do you see a way of applying the ideas of this paper to such methods?

**Limitations:**

The only limitation I see is the reduced interpretability/editability of the model compared to other approaches, but this is addressed by the authors. I see no issues with societal impact beyond generic concerns that are relevant to all scene modelling methods.

---

> ### Author Rebuttal · Authors · 2024-08-06
>
> >**Applying the idea in 3D Gaussian Splatting**
>
> Thank you for your interesting suggestion. Recently, 3DGS techniques are advancing rapidly and there are some concurrent works trying to reconstruct the surface [1*] or render reflections [2*]. GaussianShader [2*] introduces shading attributes for each 3D Gaussian, such as diffuse, tint, roughness and normal, to model reflections. Its rendering is similar to Ref-NeRF. Therefore, we think it is possible to apply our idea to 3DGS. We can combine the original 3DGS and GaussianShader representation with an optimizable “weight” attribute, similar to the weight field in our method, that is assigned to each 3D Gaussian point. We consider this as a potential future work.
>
> >**Reduced editability**
>
> We discuss this limitation in the paper (Sec. G). Though our method does not support editability, with the high-quality mesh extracted from our method, it is possible to perform editing tasks such as relighting. For example, recent single image relighting methods [3*] use diffusion models for relighting and adopt “radiance cues” -- renderings of the object’s geometry with various roughness levels under the target environment illumination -- as conditioning. The mesh extracted from our method can be used to render such radiance cues.
>
> [1*] Huang et al. 2D Gaussian Splatting for Geometrically Accurate Radiance Fields.  SIGGRAPH, 2024
>
> [2*] Jiang et al. GaussianShader: 3D Gaussian Splatting with Shading Functions for Reflective Surfaces. CVPR 2024
>
> [3*] Zeng et al. DiLightNet: Fine-grained Lighting Control for Diffusion-based Image Generation. SIGGRAPH, 2024

---

### Author Rebuttal · Authors · 2024-08-06

We thank the reviewers for their valuable feedback. In this global rebuttal, we address the common questions raised by the reviewers as follows:

> **Physical interpretation**

In this work, we mainly focus on robust surface reconstruction of real-world complex scenes with both reflective and non reflective surfaces. Current NeRF methods designed to handle reflections are mainly evaluated on synthetic datasets or scenes captured in well-controlled environments. It is challenging but important to extend them to more general scenes.

**Intuition on the need of a dual representation.**
To render reflective surfaces, Ref-NeRF [43] separates the color into physical components such as diffuse and specular components. During our extensive experiments, we find that this representation is not robust in complex real-world scenarios. As mentioned as a limitation in the Ref-NeRF paper, the representation does not “explicitly model inter-reflections or non-distant illumination”. Therefore, complex reflections that we can find in real world scenes will not be handled by similar methods.  As discussed in L157-169, we experimentally observe advantages and limitations using existing camera view and reflected view radiance fields separately.  Thus we decide to exploit the advantages of two radiance fields by combining them with a learnable weight.

**Ambiguous interpretation of reflections as geometry.**
Our custom baseline “RefV” is a physically-based model, where we follow Ref-NeRF and separate the color into diffuse and specular components. We observe that accurately separating diffuse and specular components using RefV only is a difficult task, as shown in Fig.13. In particular, view dependent effects can be represented as either geometry artifacts (holes and bumps) or with specularity. We discuss this problem in L533-543.

**Ambiguous interpretation of reflections as diffuse colors.**
Moreover, diffuse reflected components can be represented as both diffuse color and specular reflections due to the low frequency.  We observe that separating reflections and color is more challenging when the distant reflected world is less detailed. For example, the gardensphere scene is easier to reconstruct with RefV only as the background (trees, buildings) is clearly reflected. On the other hand, the sedan scene is more challenging as the reflections of the background on the car are more blurry. This observation coincides with the limitations listed in the Ref-NeRF paper.

**Ambiguous interpretation of diffuse colors as reflections.**
We find that even if the color decomposition like Ref-NeRF does not fail, the color components may not be accurate enough. For example, BakedSDF [50] follows Ref-NeRF and uses diffuse and specular color components. However, we find that BakedSDF does not accurately separate these two components. As shown in Fig.1 of the rebuttal PDF, the specular component wrongly represents the diffuse colors of the objects that have almost no reflections, such as leaves and grass.

Due to the many limitations of physics based models for real world scenes, we propose to extend such methods by unifying a physics based radiance field with an additional camera view radiance field. Therefore leading to more robust reconstructions, as highlighted by reviewers.

---

### Decision · Program_Chairs · 2024-09-25

**Decision:**

Accept (poster)

**Comment:**

The paper describes a specific self-contained improvement to NeRF: instead of using the camera direction only as in classic NeRF and instesd of using the reflected direction in combination with some "proto-shading" information as in RefNeRF, the proposal is to simply use both and learn how to blend between them. The idea is not extremely tight to NeRF, it could also be applied to other view-dependent appearance.

All reviewers understood and liked the idea, result and exposition.

The downside is, that the decomposition does not in the end provide any new insight or result in a semantic or physical separation.

The rebuttal admitted this limitation, providing some discussion and was generally well received to also clarify some details.

The AC read and and understood the paper in detail, certifies the reviews and the dsicussion were conducted orderly and hence can only recommend acceptance.